# Preliminary Evaluation of a Novel Microwave-Assisted Induction Heating (MAIH) System on White Shrimp Cooking

**DOI:** 10.3390/foods10030545

**Published:** 2021-03-06

**Authors:** Yi-Chen Lee, Chih-Ying Lin, Cheng-I Wei, Hung-Nan Tung, Kuohsun Chiu, Yung-Hsiang Tsai

**Affiliations:** 1Department of Seafood Science, National Kaohsiung University of Science and Technology, Kaohsiung 811213, Taiwan; 1021234101@stu.nkmu.edu.tw; 2Department of Nutrition and Food Science, University of Maryland, College Park, MD 20742, USA; wei@umd.edu; 3Can Pack Commercial Co., LTD, & Bottle-Top Machinery Co., LTD, Taichung 407, Taiwan; nan@bio-promotion.com.tw or; 4Department of Aquaculture, National Kaohsiung University of Science and Technology, Kaohsiung 811213, Taiwan; kuohsun@nkust.edu.tw

**Keywords:** microwave, induction heating, white shrimp, food quality

## Abstract

The microwave-assisted induction heating (MAIH) system provides comprehensive heating by combining microwave heating (with 1300 W of power and 2450 MHz of frequency) in the top part and induction heating (with 1800 W of power) in the bottom part. In this study, fresh white shrimps were placed in a sealed crystallized polyethylene terephthalate (CPET) container and heated in the MAIH system at two temperatures (130 and 90 °C) from 60 to 120 s. Afterwards, the shrimp samples were rapidly cooled, and the changes in the shrimp quality, including the appearance, cook loss, aerobic plate count (APC), color values, and texture, during the heating process were analyzed. The results demonstrate that longer heating times decrease the APC levels, but increase the cook loss, color values (lightness, redness, and whiteness), and texture (hardness, cohesiveness and chewiness) of the white shrimp samples. In particular, the white shrimp is fully cooked and gains a completely red appearance, along with no APC detected after heating in the MAIH system at 130 °C for at least 80 s or at 90 °C for at least 100 s. In summary, to achieve a good appearance, no APC detected, and low cook loss, the following heating conditions are recommended for cooking white shrimp in the MAIH system: heating at 130 °C for 80 s or at 90 °C for 100 s. This novel MAIH technology allows food to be heated and sterilized after being packed, thereby eliminating the post-pollution issue. To the best of the authors’ knowledge, this is the first study to evaluate the use of MAIH in the application of food processing.

## 1. Introduction

Microwave is a form of electromagnetic radiation, with frequencies ranging from 300 to 300 GHz, where the corresponding wavelengths range from 1 m to 1 mm [1]. According to the US Federal Communications Commission (FCC), the microwave frequencies used in general industry and household (commercial) applications are 915 and 2450 MHz, respectively [2]. Similar to light, microwave is transported in a straight line and is reflected upon hitting a metal plate. Microwaves can be absorbed by water or water-containing food to generate heat. However, microwaves cannot be absorbed by air, glass, pottery, polyethylene, and polypropylene [2,3]. When a food is placed in a microwave electric field, the microwave will induce dipole vibration and rotation on the polar molecules contained in the food (e.g., water). These intense molecular movements then result in heat generation in the food. The major difference between microwave heating and traditional heating is how heat is supplied to the food. In traditional heating methods, the thermal energy is transferred from the surface to the interior of the food, which results in a slow heating rate. In microwave heating, however, the food is heated directly from the inside. Therefore, microwave heating has several advantages, including a short heating time, a rapid temperature rise, and less damage to food nutrients and flavors [3]. Furthermore, microwave ovens are generally easy to operate and have a low maintenance cost. As a consequence, microwave heating has been widely used in the field of food processing, such as in microwave reheating of food, microwave thawing, and microwave drying [4].

Nevertheless, one primary drawback of microwave heating is the non-uniform temperature distribution during the heating cycle, which can often lead to the formation of local cold-spots or hot-spots in the food [2]. This temperature non-uniformity can cause severe issues in terms of food quality and safety, which limits the commercial use of microwave heating to only a few applications [5]. To overcome this drawback, recent studies have proposed the concept of hybrid/combination mode, in which the microwave method is combined with the traditional method to heat food [6]. This combination is also known as microwave-assisted (MA) processing technology. The MA combination technology not only incorporates the advantages of microwaves for heating food, but also overcomes other shortcomings of traditional food processing technologies. MA processing technology has been shown to provide energy savings, improved product quality, and reduced processing time and cost [6]. Therefore, this method has been widely used in many applications including MA freeze drying, MA vacuum frying, MA ultrasonic drying, and MA infrared heating [6]. Recently, the research team at Washington State University developed a novel thermal processing technology, known as microwave-assisted thermal sterilization (MATS), to realize the commercial sterilization and pasteurization of food [7]. In MATS, the food package is immersed in pressurized hot water and subjected to 915-MHz microwave heating simultaneously to achieve rapid sterilization. The MATS system is a closed system consisting of four components (i.e., preheating, microwave heating, heat preservation, and cooling). These four components are arranged in order, and also represent four different processing steps. Each component contains a separate water circulation system that is composed of a pressurized water tank and a plate heat exchanger [6,7]. The products are conveyed by a non-metallic mesh-like conveyor belt that extends from the preheating end to the cooling end. During transportation, the product is heated by the microwaves and the pressurized circulating water simultaneously [7]. Notably, the use of water as the heating medium can resolve the issues of temperature non-uniformity and prevent the formation of cold-spots and hot-spots on the edges of foods during microwave heating [8]. MATS is also the first and the only microwave sterilization technology for low-acid food approved by the U.S. Food and Drug Administration (FDA) [9,10]. Thus far, the U.S. FDA has approved the use of MATS for processing boxed mashed potato and sterilizing salmon packed in soft bags [8,9,10].

Recently, Bottle Top Machinery Co., LTD., Taiwan, invented a microwave-assisted induction heating (MAIH) system that is a modularized hybrid microwave heating system with a detachable chamber [11] (Figure 1). The top half of the chamber comprises a microwave heating unit, while the bottom half of the chamber is an electromagnetic induction heating unit. These two parts together form a MAIH system as a result of microwave resonance cavity [11]. Before operation of this system, the fresh food ingredient to be heated is first placed into a crystallized polyethylene terephthalate (CPET) box and subsequently wrapped with a sealing film. Then, the box is placed in the induction half-cavity (formed by the top and bottom chamber with a tight fitting), where it is heated by both the microwaves and electromagnetic induction. The MAIH system can reduce the heating time significantly and resolve the drawbacks of the conventional microwave heating method and provide uniform heating of food. In addition, the heating and sterilization processes are conducted simultaneously in one step, while no man-made packaging is required in the follow-up procedure, which reduces the pollution issue. Therefore, the MAIH heating system allows food to be preserved over a long period [11].

White shrimp is one of the most commonly consumed shrimps in the world, one half of which is consumed in processed form, largely frozen [12]. There is no information on the application for food processed by MAIH, especially white shrimp. Therefore, the main aim of this study was to develop a small-scale MAIH system heating process for pre-packaged white shrimp in saline and evaluate the quality attributes of this product, including the appearance, color, cook loss, aerobic plate count, and texture. This study should provide useful information for future commercial MAIH applications in seafood processing.

## 2. Materials and Methods

### 2.1. Sample Preparation

Live white shrimps (*Litopenaeus vannamei*) were purchased from the Fishery Market in Kaohsiung city, Taiwan, wrapped in crushed ice, and transported to the laboratory. After arriving at the laboratory, these shrimps were then washed with tap water and mixed with 1% saline in a 1:1 (*w/v*) ratio in a white CPET container. A total of 100 g shrimp (15–18 g/each shrimp) was filled into each CPET container (6.5 cm i.d. × 3.0 cm height). Finally, the container was sealed with a polyethylene terephthalate (PET) film for use in the subsequent analysis.

### 2.2. MAIH Processing

The microwave-assisted processing of pre-packaged shrimps was performed in a small-scale MAIH system developed at Bottle Top Machinery Co., LTD., Taiwan (Figure 1). Before processing, the sealed CPET container containing the white shrimp was placed in the induction half-cavity. The power and frequency of the microwave heating unit in the MAIH system were set to 1300 W and 2450 MHz, respectively. The heating temperature of the electromagnetic induction heating unit (with 1800 W power) was set to two temperatures at 130 and 90 °C for the test. At the heating temperature of 130 °C, the heating time was varied as 60, 70, 80, 90, and 100 s; at the heating temperature of 90 °C, the heating time was varied as 60, 70, 80, 90, 100, and 120 s. For each heating time in the test conditions, the MAIH system was only used for induction heating (microwave turned off) in the final 30 s (Table 1). When the heating process was completed, the induction half-cavity was removed from the MAIH system and cooled in ice water for 6 min. Subsequently, the CPET container was taken out from the induction half-cavity, and the heated shrimp samples were removed from the container and analyzed by the following tests.

### 2.3. Cook Loss Measurement

After heating and cooling, the shrimps were drained on the stainless-steel screen for 6 min. The cook loss was calculated by first dividing the change in the shrimp weight before and after heating by the original weight of the white shrimp, and then multiplying the result by 100 [13].
Cook loss = [(weight of raw sample−weight of cooked sample)/(weight of raw sample)] × 100(1)

### 2.4. Determination of Aerobic Plate Count (APC)

The minced samples (25 g) were placed in a sterilize blender (containing 225 mL of 0.85% sterilized saline). The mixture was then homogenized at a 1200-rpm speed for 2 min and serially diluted with sterilized saline for 1:10 (*v/v*) dilutions. Subsequently, 0.1 mL of the dilutes was spread on aerobic plate count (APC) agar (Difco, BD, Sparks, MD, USA) with 0.5% NaCl, and then the Petri dish containing the growth medium was incubated at 30 °C for 24–48 h. Finally, the bacterial colonies grown on the plate were counted and expressed as log_10_ colony-forming units (CFUs) per gram [13].

### 2.5. Appearance and Color Measurement

The appearance of the shrimp samples was recorded on a white background by using an SLR camera (EOS 60D, Canon Inc., Japan). The CIE (Commission internationale de l′eclairage) color values of peeled shrimp after heating in the MAIH system at two temperatures (130 and 90 °C) were analyzed in six replicate measurements by using a colorimeter (NE-4000, Nippon Denshoko, Japan). The *L**, *a**, *b** (*L** (lightness), *a** (+a, red, ‒a, green), and *b** (+b, yellow; ‒b, blue)) values were displayed on the screen of the colorimeter. The *W* value (whiteness) of the peeled shrimp was then calculated by applying the following equation, and the average value was taken as the final result [14].
Whiteness = 100 − [(100 − L*)^2^ + a*^2^ + b*^2^]^1/2^(2)

### 2.6. Texture Measurement

In order to evaluate changes in the texture of shrimp after MAIH processing, a texture profile analysis (TPA) was performed on the cooked shrimp meat using a CT3 Texture Analyzer (Brookfield Engineering Laboratories, Middleboro, MA, USA). The TPA includes measurement of the hardness, cohesiveness, springiness, and chewiness. The test conditions were set as follows: probe: TA39 cylinder, 2 mm D, 20 mm L; target value: 4.00 mm; predicted speed: 2 mm/s; test speed and return speed: 1.5 mm/s; load at trigger point: 2 g. The samples were analyzed in triplicate, and the average values were calculated. Four shrimps were tested from each treatment (*n* = 4), with measurements taken at four different locations on each shrimp. The four locations were located in the front four sections of the peeled shrimp body.

### 2.7. Sensory Evaluation

To assess consumer acceptance of the novel MAIH method applied for the cooking of shrimp, sensory analysis of the samples treated with MAIH-130 °C for 80 s and MAIH-90 °C for 100 s was evaluated in comparison with conventional boiled shrimps. In the boiled shrimp treatment, the shrimps were heated in a thermostat-controlled water bath (40 × 30 × 20 cm^3^) containing around 6 L of 1% saline, with temperature at 90 ℃ for 130 s, after which samples were packed in a Ziploc bag (S.C. Johnson & Son, Racine, WI, USA) and inserted into an ice bath for 5 min. Sensory evaluation of the boiled shrimps and the shrimps cooked using MAIH-130 °C or 80 s and MAIH-90 °C for 100 s was undertaken by 25 untrained panelists from the Department of Seafood Science, National Kaohsiung University of Science and Technology, aged 20–26, using a 9-point hedonic scale, where 9: like extremely; 7: like moderately; 5: neither like or nor dislike; 3: dislike moderately; 1: dislike extremely [15]. Panelists were regular consumers of shrimp and had no allergies to shrimp. All the panelists were asked to evaluate for color, odor, texture, flavor, taste, and overall acceptance. Samples were presented in plates coded with three-digit random numbers. Panelists were asked to rinse their mouths using drinking water between samples.

### 2.8. Statistical Analysis

All the measurement data were analyzed using the Statistical Package for Social Sciences (SPSS) version 16.0 (SPSS Inc., Chicago, Il, USA). One-way analysis of variance (ANOVA) and LS (Least Squares) means Tukey HSD (Honestly Significant Difference) were used to determine whether there were any statistically significant differences between the measurement data. A *p* value less than 0.05 indicated a significant difference.

## 3. Results and Discussion

### 3.1. Appearance of Shrimp after Heating in MAIH System

The appearance of the shrimp samples after heating in the MAIH system at 130 °C for 60‒100 s and at 90 °C for 60‒120 s is shown in Figure 2. The shrimp heated at 130 °C for 60‒70 s had a red appearance, but the head and legs remained dark brown. This color profile indicates that the shrimp were not cooked well. After heating above 80 s, the entire shrimp body became red and fully cooked. However, after heating for 100 s, the shrimp shell became dry, flat, and detached from the shrimp meat, indicating that the shrimp was overcooked (Figure 2A). Upon heating at 90  C for 60 s, the shrimp body remained dark brown. When the heating process was extended to 70–90 s, the color of the shrimp turned red, while the head and legs remained dark brown, which indicates that the shrimp was not fully cooked. At heating for 100 s, the entire shrimp body turned red and was fully cooked. However, the white shrimp was overcooked when heated for 120 s, where the shrimp shell became dry, flat, and detached from the shrimp meat (Figure 2AB).

### 3.2. APC and Cook Loss of Shrimp Heated in MAIH System

Table 2 shows the APC of the shrimp sample after heating via MAIH at 130 and 90 °C. Initially, the raw white shrimp had 6.24 log CFU/g of APC. When heated at 130 °C, the APC of the shrimp was found to decrease with increasing heating time. The APC of the sample was nondetectable (<2.0 log CFU/g) after heating above 80 s. When heated at 90 °C, the APC of the shrimp was also found to decrease with increasing heating time. The APC of the sample became negligible (<2.0 log CFU/g) after heating for 100 and 120 s (Table 2). Therefore, at the heating temperatures of 130 and 90 °C, the processing time in the MAIH system must be set to at least 80 and 100 s, respectively.

The cook loss in the shrimp samples after heating in the MAIH system at 130 and 90 °C is summarized in Table 2. Cook loss can be calculated using Equation (1). When heated at 130 °C, the cook loss in the shrimp was found to increase with increasing heating time. Specifically, the cook loss increased from 1.15% to 4.90% when the heating time was increased from 60 to 100 s. This trend indicates that prolonged heating will accelerate the loss of water from the shrimp sample. In addition, as shown by comparison with Figure 2A, the shrimp shell became dry, flat, and detached from the meat when the cook loss reached 4.90% (heating at 130 °C for 100 s). Similar features were observed at a heating temperature of 90 °C, where the cook loss also increased with increasing heating time. Specifically, the cook loss of the shrimp increased from 0.80% to 3.90% as the heating time increased from 60 to 120 s. Furthermore, as shown by comparison with Figure 2B, the shrimp shell became dry, flat, and detached from the meat when the cook loss reached 3.90% (heating at 90 °C for 120 s). Thermal processing causes denaturation of muscle protein, which is the main mechanism that results in moisture loss [14]. Erdogdu et al. [16] proposed that the cook loss in white shrimp is primarily attributed to the contraction of myofibril protein and the denaturation of collagen during the heating process. Most of the water contained in the muscle is located in the muscle fibers between the thick and thin filaments. During the heating process, contraction and hardening of the muscle tissue causes the internal pressure to change, which further allows the water to be released from the muscle structure. Therefore, the weight of the white shrimp is reduced after heating, which in turn changes the overall density and volume of the sample [17]. The results of this study are in agreement with a previous study report, in which the cook loss in salmon increased significantly with increasing temperature and heating time, and the cook losses reached an equilibrium, which was 13.2%, 15.4%, 18.5%, and 20.2% for 100, 111.1, 121.1, and 131.1 °C, respectively [17,18]. However, more than 30% cook loss in blue mussel occurred at pasteurization temperatures (65 to 90 °C) [14]. Erdogdu et al. [16] stated that the cook loss in white shrimp ranged from 15.8% to 28.7% in boiling water at 90 °C and increased with decreasing size of samples. The higher cook loss in white shrimp samples reported by Erdogdu et al. [16] compared to this study is probably due to the longer cooking time (>3 min), as well as differences in cooking processes.

### 3.3. Color of Shrimp Heated in MAIH System

Color is one of the main sensory characteristics used to determine the quality and acceptability of food. It is also used by consumers as a reference indicator to evaluate products. In other words, the color of seafood is very important in terms of consumer perception of seafood quality and is a dominant factor in consumer purchasing decisions [19]. Crustaceans such as crabs and shrimps contain a red carotenoid known as astaxanthin. This substance binds to proteins in the bodies of live crustaceans to form crustacyanin, which appears brown or dark brown. During the heating process, the protein becomes denatured and releases astaxanthin (red color), which changes the color of the shell and flesh to red [20]. Table 3 shows the change in the color of the peeled shrimp after the heating process in the MAIH system at 130 and 90 °C. Initially, the *L**, *a*, b**, and *W* values for the raw peeled shrimp are 25.23, 1.90, −1.06, and 25.20, respectively. The levels of *L** in heated samples increased with the increase in heating time, regardless of temperature at 130 or 90 °C. Under the heating process at 130 °C, the *L** values significantly increased from 41.53 in the 60 s heating sample to 52.01 in the 100 s heating sample (*p* < 0.05). Similarly, the *L** values of the heated samples at 90 °C increased from 28.22 for 60 s heating to 52.57 for 120 s heating (*p* < 0.05). *L** represents the lightness and a greater *L** indicates a lighter appearance of the shrimp sample. With regard to *a** (redness), in heated samples at 130 °C, the values increased markedly from 6.44 for 60 s heating to 11.73 for 100 s heating (*p* < 0.05). In samples heated at 90 °C, the *a** values first increased from 2.71 at 60 s to 15.46 at 100 s, but then dropped to 10.73 at 120 s (*p* < 0.05). This trend suggests that prolonged heating will cause the pigment to decompose and the color to fade. However, the *b** (yellowness) values for heated samples were found to decrease slightly with increasing heating time. After heating at 130 and 90 °C, the *b** values ranged between −1.45 and −2.55 and between −1.01 and −1.86, respectively. Whiteness (*W*) can be calculated using Equation (2). The *W* values increased from 41.15 at 60 s to 50.52 at 100 s after heating at a temperature of 130 °C, and from 28.16 at 60 s to 51.33 at 120 s after heating at a temperature of 90 °C (*p* < 0.05). These results indicate that the appearance of the peeled shrimp becomes increasingly white with increasing heating time. This is due to the denaturation of heme proteins (e.g., hemoglobin and myoglobin), and the whitening phase is accompanied by the production of the cooked seafood meat hemoprotein [17].

In summary, regardless of the heating temperature, the color parameters (*L**, *a**, and *W*) of the shrimp body were found to increase with increasing heating time. According to the color information shown in Table 3 and the appearance depicted in Figure 2, the white shrimp is fully cooked after heating for 80 s at 130 °C or heating for 100 s at 90 °C. At this stage, the white shrimp gains a completely red color.

### 3.4. Texture of Shrimp Heated in MAIH System

Figure 3 shows the changes in the texture of the peeled shrimp after heating via MAIH at 130 and 90 °C for different durations. The textural properties include the hardness, cohesiveness, springiness, and chewiness. Hardness refers to the amount of force exhibited in the first bite at maximum compression. In this study, a texture profile analyzer was used to measure the critical weight required to induce a certain level of deformation of the peeled shrimp. Initially, the hardness of the raw peeled shrimp was measured to be around 100 (g). During the heating process at 130 °C, the average hardness of the heated samples was found to increase and ranged from 260 to 300 (g). However, no significant difference was observed between the different heating times (*p* > 0.05) (Figure 3A. Similarly, the average hardness of the heated samples was also found to increase with in-creasing heating time at 90 °C. Nevertheless, no statistically significant difference (*p* > 0.05) was observed between the heating times of 80 and 120 s, as the hardness ranged from 260 to 310 (g) (Figure 3A).

Cohesiveness describes the internal adhesion force of the food. When the probe of the texture profile analyzer is brought into full contact with the sample, the probe can remain clean without any adhered sample pieces. In this case, the cohesiveness is defined as the ratio between the first compression area and the second compression area. The cohesiveness of the raw peeled shrimp is around 0.22. During the heating process at 130 °C, the average cohesiveness of the heated sample was found to increase with increasing heating time and reached a maximum value of around 0.75 after heating for 100 s. A significant difference was observed between the different times (*p* < 0.05) (Figure 3B). During the heating process at 90 °C, the average cohesiveness of the heated samples was found to increase with increasing heating time and reached a maximum value of around 0.62 after heating for 100 s. However, the cohesiveness decreased to 0.58 when the heating time was further increased to 120 s, but no significant difference (*p* > 0.05) in cohesiveness values between 100 and 120 s samples was found (Figure 3B).

Springiness is used to characterize the restoration ratio in height or volume of a deformed sample after the deformation load is removed (i.e., the sample is subjected to the same initial condition before the load is applied). The average springiness ranged from 5.0 to 5.5 for the raw shrimp samples and those processed at 130 or 90 °C for different heating times. No statistically significant difference was observed between the different heating samples (*p* > 0.05) (Figure 3C).

Chewiness is the energy (mj) required to chew solid food until it can be swallowed. The chewiness of raw peeled shrimp is around 5.9 (mj). During the heating process at 130 °C, the chewiness of the heated sample was found to increase with increasing heating time (*p* < 0.05) and reached a maximum value of 12.2 (mj) after heating for 100 s. During the heating process at 90 °C, the chewiness of the samples also increased with increasing heating time (*p* < 0.05) and reached a maximum value of 12.3 (mj) after heating for 120 s. Notably, the chewiness of the samples heated at 130 °C for 80 s was the same as that of the samples heated at 90 °C for 70–90 s. In general, among the textural properties, the hardness, cohesiveness, and chewiness of the shrimp sample all tend to increase with increasing heating time, regardless of the heating temperature. It is inferred that these changes can be attributed to the denaturation of the muscle protein in the shrimp during the heating process, which causes contraction of the tissue structure and the release of water from the shrimp [16]. As a consequence, the texture of the tissues becomes harder [16]. A similar finding was also reported by Bhattacharya et al. [21], who found that the texture (i.e., hardness) of Pacific chum salmon (*Oncorhynchus keta*) increased as the heating temperature and time increased using hydrothermal processing. However, Yagiz et al. [22] demonstrated that cooked salmon samples had lower hardness, cohesiveness, and chewiness values compared to control samples. Kong et al. [17] also reported that the shear force of salmon (*O. gorbuscha*) reduced with increasing temperature. The difference may be due to the counteraction of hardening and softening reactions, while high temperatures resulted in rapid disintegration and fragmentation of the fish meat, promoting the reduction of shear force [17].

### 3.5. Sensory Properties of Cooked White Shrimp with MAIH Treatments

The sensory properties of white shrimp subjected to various treatments (the boiled shrimps, and the shrimps cooked using MAIH-130 °C and MAIH-90 °C) are presented in Table 4. The scores of color, odor, texture, flavor, taste, and overall acceptance in all differently treated samples ranged from 7.71 to 7.85, 7.83 to 7.95, 7.73 to 7.78, 7.43 to 7.58, 7.34 to 7.42, and 7.52 to 7.60, respectively. No differences in sensory scores for all attributes were noticeable between the boiled shrimps and those cooked using MAIH-130 °C and MAIH-90 °C (*p* > 0.05). Therefore, the white shrimps cooked with the MAIH system had good sensory properties as compared with the boiled shrimps.

In microwave-assisted food processing technologies, MATS is an emerging technology that is an efficient sterilization method to inactivate food bacterial load, along with reducing the degradation of nutrient and flavors in packaged food [7]. It is a closed system including four sections, i.e., preheating, microwave heating, holding, and cooling, while each section has a separate water circulation system that is composed of a pressurized tank and plate heat exchanger [8]. Nevertheless, the cost of a MATS system for the regular processing line is a problem for food plants. In this study, the MAIH system consisting of microwave and induction heating is a novel, simple, fast, and cheap piece of equipment for food thermal processing. Although there is little research on this MAIH, it has great potential for developing short-time in-package sterilization and pasteurization processes in the food industry. Recently, Fan et al. [23] reported that higher cooking uniformity and lower average overheating degree for crayfish tail heated by microwave (MW) were observed, as compared to boiling water heating (BW), but the central overheating degree of crayfish tail heated by MW was higher than by BW, and the overheating might be considered as a more important cause of the texture deterioration of crayfish tails heated by MW than by BW. Therefore, further studies on the MAIH are required to elucidate system validation and shelf-life of food using MAIH.

As shown in the analyses discussed above, the white shrimp is fully cooked after heating via MAIH at 130 °C for 80 or 90 s, or at 90 °C for 100 s. At this stage, the shrimp sample gains a completely red appearance and has a negligible bacterial load. In addition, the shrimp sample is subjected to minor cook loss while possessing decent textural qualities, including good hardness, cohesiveness, and chewiness. Therefore, the recommended heating condition for cooking white shrimp in the MAIH system is either 130 °C for at least 80 s or 90 °C for at least 100 s. This MAIH technique has the potential to heat rapidly, reduce heating time, and cook and sterilize simultaneously in white shrimps, but further work should be conducted to compare the corresponding results, concerning cooking methods when shrimp is subjected to electromagnetic induction heating only, without microwave heating.

## 4. Conclusion

In this study, white shrimp samples were heated in the MAIH system at 130 and 90 °C. With increasing heating time, the appearance of the white shrimp indicates that it is fully cooked or even overcooked. A longer heating time leads to increasing cook loss, color difference (*L**, *a**, and *W*), and texture (hardness, cohesiveness, and chewiness) of the shrimp, but a decreasing APC level. Therefore, it is recommended to heat the white shrimp in the MAIH system at 90 °C for at least 100 s or at 130 °C for at least 80 s. These heating conditions confer an appreciable appearance, nondetectable APC, and relatively low cook loss for the shrimp sample. However, heating the shrimp for too long leads to overcooking, where the shrimp shell becomes dry, flat, and detached from the shrimp meat. In this study, we evaluated the preliminary application of a novel food processing technology, MAIH, for heating white shrimp, that could offer several advantages, including rapid heating and the ability to cook and sterilize simultaneously.

## Figures and Tables

**Figure 1 foods-10-00545-f001:**
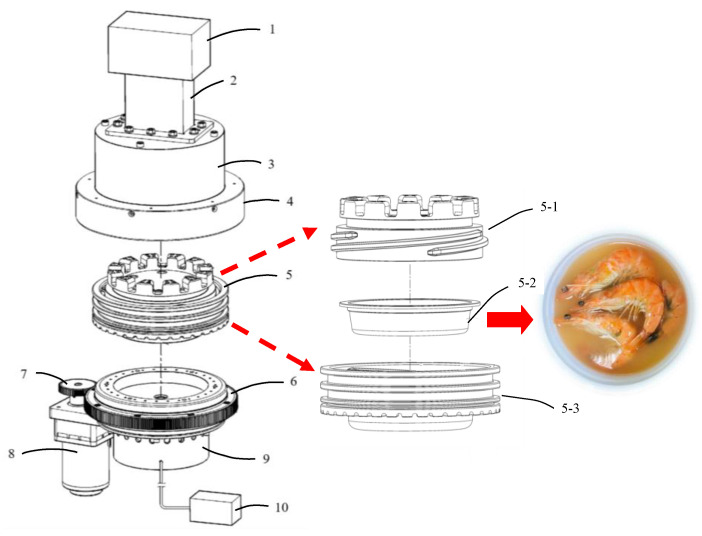
Microwave-assisted induction heating (MAIH) system diagram developed by Bottle Top Machinery Co., LTD., Taiwan. 1: Microwave heating unit; 2: Waveguide; 3: Microwave half-cavity body; 4: Microwave half-cavity cover; 5: Induction half-cavity; 6: Revolving take-over turntable; 7: Gears; 8: Rotating motor; 9: Induction heating unit; 10: Induction heating power controller; 5-1: Induction half-cavity upper cover; 5-2: Sealing crystallized polyethylene terephthalate (CPET) container; 5-3: Induction half-cavity body.

**Figure 2 foods-10-00545-f002:**
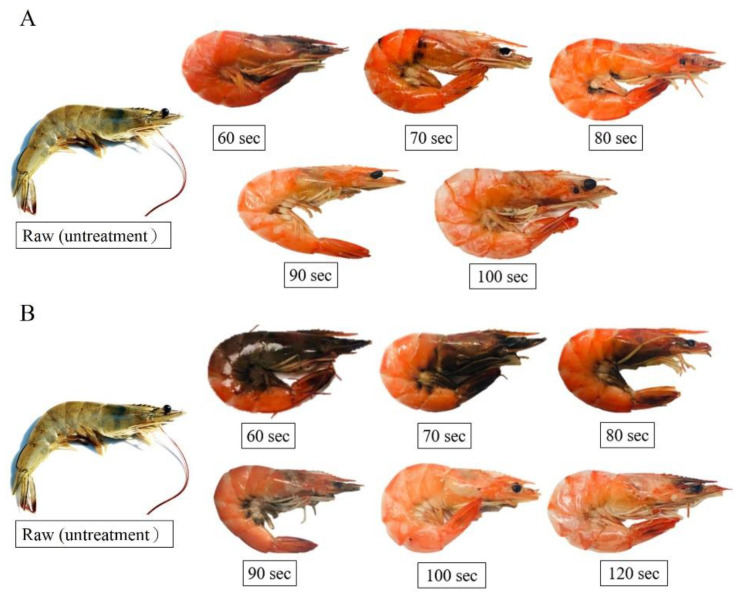
Appearance changes in white shrimp using microwave-assisted induction heating (MAIH) system set at 130 °C (**A**) and 90 °C (**B**) for different heating times.

**Figure 3 foods-10-00545-f003:**
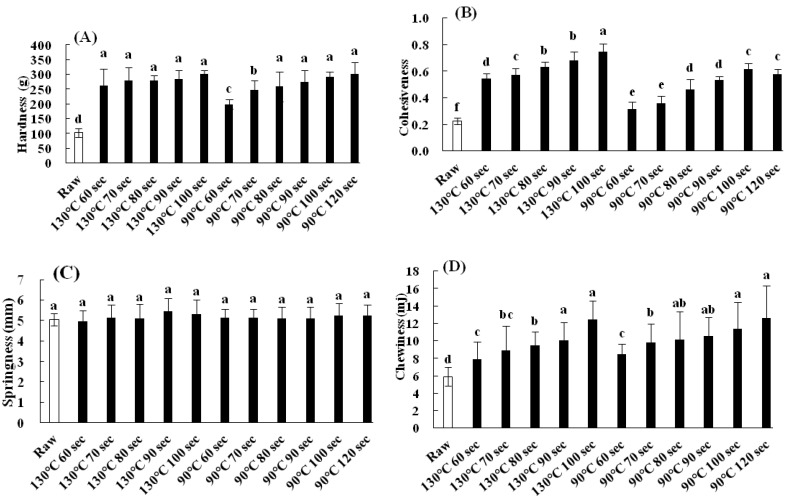
Changes in texture properties of hardness (**A**), cohesiveness (**B**), springiness (**C**), and chewiness (**D**), in peeled white shrimp using microwave-assisted induction heating (MAIH) system set at 130 and 90 °C for different heating times. Each value is the mean ± standard deviation (*n* = 4); the different letters by the bars indicate significant differences (*p* < 0.05).

**Table 1 foods-10-00545-t001:** Heating conditions of MAIH system for white shrimp cooking.

Temperature	First Stage Heating (s)	Second Stage Heating (s)	Total Heating Time (s)
	MW + IH	IH	
130 °C	30	30	60
	40	30	70
	50	30	80
	60	30	90
	70	30	100
90 °C	30	30	60
	40	30	70
	50	30	80
	60	30	90
	70	30	100
	90	30	120

MW + IH: Microwave-assisted heating + induction heating; IH: Induction heating.

**Table 2 foods-10-00545-t002:** Changes in aerobic plate count (APC) and cook loss in white shrimp using microwave-assisted induction heating (MAIH) system set at 130 and 90 °C for different heating times.

Treatments	Heating Time (s)	APC (log CFU/g)	Cook Loss (%)
Raw shrimp		6.24 ± 0.02 ^a^	0
130 °C	60	3.47 ± 0.18 ^c^	1.15 ± 0.07 ^de^
	70	3.23 ± 0.12 ^c^	1.35 ± 0.07 ^de^
	80	<2.0 ^e^	2.55 ± 0.21^c^
	90	<2.0 ^e^	3.45 ± 0.49 ^b^
	100	<2.0 ^e^	4.90 ± 0.42 ^a^

90 °C	60	4.28 ± 0.06 ^b^	0.80 ± 0.14 ^e^
	70	4.16 ± 0.09 ^b^	1.50 ± 0.28 ^d^
	80	3.30 ± 0.18 ^c^	2.20 ± 0.14 ^c^
	90	2.65 ± 0.07 ^d^	2.65 ± 0.21 ^c^
	100	<2.0 ^e^	3.35 ± 0.07 ^b^
	120	<2.0 ^e^	3.90 ± 0.28 ^b^

Each value is the mean ± standard deviation (*n* = 3); the different letters in the same column indicate significant differences (*p* < 0.05).

**Table 3 foods-10-00545-t003:** Changes in color (*L**, *a**, *b**, *W* value) in peeled white shrimp using microwave-assisted induction heating (MAIH) system set at 130 and 90 °C for different heating times.

Treatments	Heating Time(s)	*L**	*a**	*b**	*W*
Peeled raw shrimp		25.23 ± 0.60 ^i^	1.90 ± 0.12 ^g^	−1.06±0.04 ^a^	25.20±0.58^i^
130 °C	60	41.53 ± 0.91 ^f^	6.44 ± 0.50 ^d^	−1.45 ± 0.18 ^bc^	41.15 ± 0.89 ^f^
	70	45.45 ± 1.08 ^e^	6.67 ± 0.47 ^d^	−1.94 ± 0.19 ^d^	45.01 ± 1.09 ^e^
	80	49.49 ± 0.58 ^c^	10.28 ± 0.44 ^c^	−2.38 ± 0.23 ^e^	48.40 ± 0.55 ^cd^
	90	49.65 ± 0.48 ^c^	10.45 ± 0.45 ^c^	−2.54 ± 0.46 ^e^	48.51 ± 0.45 ^cd^
	100	52.01 ± 1.44 ^ab^	11.73 ± 1.12 ^b^	−2.55 ± 0.30 ^e^	50.52 ± 1.23 ^b^

90 °C	60	28.22 ± 0.85 ^h^	2.71 ± 0.37 ^f^	−1.01 ± 0.71 ^a^	28.16 ± 0.84 ^h^
	70	37.55 ± 1.06 ^g^	3.97 ± 0.10 ^e^	−1.37 ± 0.13 ^b^	37.40 ± 1.06 ^g^
	80	42.07 ± 0.65 ^f^	4.41 ± 0.28 ^e^	−1.66 ± 0.25 ^cd^	41.87 ± 0.64 ^f^
	90	48.27 ± 1.06 ^d^	6.41 ± 0.69 ^d^	−1.84 ± 0.10 ^d^	47.83 ± 1.06 ^d^
	100	51.49 ± 0.85 ^b^	15.46 ± 0.56 ^a^	−1.35 ± 0.08 ^b^	49.07 ± 0.89 ^c^
	120	52.57 ± 0.78 ^a^	10.73 ± 0.44 ^c^	−1.86 ± 0.06 ^d^	51.33 ± 0.73 ^a^

Each value is the mean ± standard deviation (*n* = 3); the different letters in the same column indicate significant differences (*p* < 0.05).

**Table 4 foods-10-00545-t004:** Sensory evaluation of white shrimps treated with boiling water at 90 °C for 130 s, MAIH at 130 °C for 80 s, and MAIH at 90 °C for 100 s.

Treatments	Color	Odor	Texture	Flavor	Taste	Overall Acceptance
Boiling water	7.85 ± 0.96 ^a^	7.83 ± 0.69 ^a^	7.78 ± 0.79 ^a^	7.43 ± 0.77 ^a^	7.38 ± 0.83 ^a^	7.60 ± 0.79 ^a^
MAIH 130 °C	7.71 ± 0.90 ^a^	7.95 ± 0.84 ^a^	7.74 ± 0.93 ^a^	7.58 ± 0.90 ^a^	7.34 ± 0.96 ^a^	7.52 ± 0.83 ^a^
MAIH 90 °C	7.78 ± 0.88 ^a^	7.79 ± 0.98 ^a^	7.73 ± 0.99 ^a^	7.55 ± 0.81^a^	7.42 ± 0.91 ^a^	7.53 ± 0.92 ^a^

Each value is the mean ± standard deviation (*n* = 25); the different letters in the same column indicate significant differences (*p* < 0.05).

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
