# Peer review of "Preliminary Evaluation of a Novel Microwave-Assisted Induction Heating (MAIH) System on White Shrimp Cooking"

_foods, 2021, doi:10.3390/foods10030545_

Round 1
Reviewer 1 Report
In its present state, this manuscript cannot be considered for publication as a full research article because the main objective is to analyse the feasibility of the Microwave-assisted Induction Heating (MAIH) from a preliminar point of view.
However, the manuscript could be considered as a short article after major revisions. This work shows some novelty with regards to the technology involved for preservation of quality of white shrimps.
The methodology used to analyze the quality of the shrimps after being treated by MAIH is correct but sensory analysis should also be included, as it represents the most important analysis for consumer acceptance. The technology involved can be great for product preservation, but if the sensory properties do not meet consumer expectations, it is not worth the study.
Therefore, the work carried out should be completed with the corresponding sensory analysis, defining the attributes and the sensory scale appropiate for white shripms.
Furthermore, in Figure 1, numbers 11, 12 and 13 appear in the legend but are not shown in the figure.
In Table 2, the parameters for determination of microbial quality should be referenced for white shrimps. In order to call it "sterilisation", it might be necessary to ensure the absence of other parameters (for example, spores) associated to shrimps.
Reviewer 2 Report
The manuscript presents interesting results concerning novel method applied for cooking of shrimp. These results might be useful for practical application. However, this manuscript requires improving due to some shortcomings. First of all reference samples should be prepared using traditional method or electromagnetic induction heating only without microwave heating. Since this was not done, the obtained results should be compared with the corresponding results, concerning cooking of shrimp, obtained by other authors.
Other comments that should be considered to make the manuscript suitable for publication:
Line 121. Probably setting the frequency of microwave heating was not possible as magnetrons usually work at approved frequency which cannot be changed.
Line 122. It is not clear what exactly the temperature of 130 and 90°C means. Is it the temperature of the heating element or the expected temperature of the processed material? Alternatively, power of electromagnetic induction heating unit can be provided.
Line 162. Pleas provide reference for determination of “W” value.
Line 178. These four locations should be clearly specified as repeatability is an important issue in the TPA test.
Line 234. Consider replacing “at” with “at least”.
Line 266. The heating systems for cooking of salmon and blue mussel should be specified. However, more appropriate option would comparing the values of cook loss taking into account shrimp cooked in traditional way as reference.
Line 304. Replace “appearance” with “color”. The same might be done in line 308.
Line 313-314. Consider using another definition of hardness referring to the maximal force at the first compression because the occurrence of breaking point may cause ambiguity.
Round 2
Reviewer 1 Report
In its current state the article is not ready yet for publication. It needs to be completed with the sensory results.
Reviewer 2 Report
The manuscript concerns important issues regarding new method of food processing. Most of the remarks was considered by the Authors. However, some issues still require addressing. First of all reference sample should be defined and prepared in an appropriate way. Also consumer acceptance should be verified by relevant assay such as sensory analysis. I recommend supplementing the manuscript with missing data or resubmiting it after it has been fully edited.
Round 3
Reviewer 1 Report
The authors have included the sensory analysis carried out to complement the physico-chemical analysis previously carried out.
In its present form the article includes the necessary analysis to come to the conclusions stated by the authors.
Reviewer 2 Report
Thank you for improving the manuscript and providing the results of the sensory analysis, including a properly prepared reference sample. In my opinion, a revised version of the manuscript can be published in its present form.
